# How does overparameterization affect features?

## Abstract

Overparameterization, the condition where models have more parameters than necessary to fit their training loss, is a crucial factor for the success of deep learning. However, the characteristics of the features learned by overparameterized networks are not well understood. In this work, we explore this question by comparing models with the same architecture but different widths. We first examine the expressivity of the features of these models, and show that the feature space of overparameterized networks cannot be spanned by concatenating many underparameterized features, and vice versa. This reveals that both overparameterized and underparameterized networks acquire some distinctive features. We then evaluate the performance of these models, and find that overparameterized networks outperform underparameterized networks, even when many of the latter are concatenated. We corroborate these findings using a VGG-16 and ResNet18 on CIFAR-10 and a Transformer on the MNLI classification dataset. Finally, we propose a toy setting to explain how overparameterized networks can learn some important features that the underparamaterized networks cannot learn.

## 1 Introduction

Overparameterized neural networks, which have more parameters than necessary to fit the training data, have achieved remarkable success in various tasks, such as image classification (He et al., 2016; Krizhevsky et al., 2017), object detection (Girshick et al., 2014; Redmon et al., 2016) or text classification (Zhang et al., 2015; Johnson & Zhang, 2016). However, the theoretical understanding of why these networks outperform underparameterized ones, which have fewer parameters and less capacity, is still limited. Several works have attempted to explain the advantages of overparametrization from various perspectives, such as the neural tangent kernel (Arora et al., 2019), the lottery ticket hypothesis (Frankle & Carbin, 2018), or the implicit regularization (Neyshabur et al., 2014). Most of these papers claim that overparameterized networks are more powerful due to their greater number of parameters. In this paper, we aim to go beyond this perspective, focusing on the case where overparameterized and underparameterized networks are compared on an *equal* footing. More specifically, we analyze their features, which are the representations learned by the hidden layers of the networks, and ensure that both overparameterized and underparameterized have the same number of features in our comparison. Analyzing neural networks features is a common in deep learning (Kornblith et al., 2021; Raghu et al., 2021; Cianfarani et al., 2022) to unveil important properties of the networks, such as generalization ability, robustness, or interpretability.

A few existing works have taken this approach to characterize overparameterized networks. Nguyen et al. (2020) studied how varying depth and width affects features and discovered a distinctive block structure in overparameterized networks. Morcos et al. (2018) observed that wider networks learn more similar features. However, these works do not study a direct comparison between overparameterized and underparameterized networks features, which could reveal if overparameterized networks capture unique features compared to underparameterized networks, and vice versa. Our paper aims to address the following question:

*Even after many concatenations of independently trained underparameterized networks, can we fully retrieve the expressive power and the performance of overparameterized networks?*

Our answer to this question is negative, and we summarize our contributions as follows:

- Section 2 introduces our feature analysis methods. Our first approach, called feature span error (FSE), uses ridge regression to measure how well overparameterized features capture underparameterized network features, and vice versa. We also introduce the feature performance (FP) method to assess feature accuracy in task performance through linear probing.

- To ensure a fair comparison between the two sets of features, we compare the features of a single overparameterized network with those concatenated from many underparameterized networks trained with different initializations. As such, both sets are of comparable size.

- Section 3 presents our numerical results. FSE and FP demonstrate that i) overparameterized and concatenations of underparameterized networks learn distinct features and ii) linear probing with the concatenated features of underparameterized networks do not reach the same test accuracy as linear probing with overparameterized features.

- We further demonstrate that the part of overparameterized features that concatenations of underparameterized networks features cannot capture, which we name overparameterized residuals, are essential in the high performance of overparameterized networks. Conversely, the underparameterized residuals do not improve the performance of a model.

- Section 4 proposes a toy setting to illustrate how overparameterized networks can learn some important features differently from underparameterized networks. Our setting partially covers the observations made in Section 3 since we do not show that overparameterized networks cannot learn some features learned by underparameterized ones. Nevertheless, it conveys crucial insights.

## RELATED WORK

**Scaling the size of neural networks.** In this paper, we construct our smaller networks by scaling down the widths of a base overparameterized network while keeping the depth fixed. Studying neural networks with scaling widths or depths has been widely done in the literature. Cybenko (1989); Hanin & Sellke (2017) analyzed the effect of varying width on the expressiveness of neural networks. On the empirical side, the effect of scaling width on the performance has been extensively studied (Zagoruyko & Komodakis, 2016; Kaplan et al., 2020) Closer to our work, (Morcos et al., 2018; Nguyen et al., 2020) study the effect of scaling width and/or depth on the features of a network. While (Nguyen et al., 2020) investigates the features across all hidden layers, (Morcos et al., 2018) demonstrates that the wider networks learn similar last layer features. We also investigate the last layer features, but with a focus on directly comparing underparameterized and overparameterized network features.

**Analyzing the features to gain insights in neural networks.** Many research has focused on analyzing and understanding the representations learned by neural networks (Merchant et al., 2020; Cianfarani et al., 2022). Many of these works provide insights about interpretability, generalization, model comparison, transfer learning, and model improvement. Utilizing similarity metrics is a common approach for analyzing features to compare different models. Nguyen et al. (2020) employ the centered kernel alignment (CKA) metric, while Morcos et al. (2018) utilize the projection weighted CCA metric to measure the similarity between features. In this paper, our focus lies on the expressivity of the features of one network compared to another, rather than solely on their similarity. To that end, we introduce the feature span error (FSE) metric, which is based on ridge regression. While using regression as a similarity metric is mentioned in (Kornblith et al., 2019), its application, to the best of our knowledge, has not been explored.

**Notations.** Throughout the paper, we use the upper case $M$ to refer to a neural network, the calligraphic $\mathcal{M}$ to refer to its features and the lower case $m$ to its feature mapping. For a matrix $\boldsymbol{A}$, we refer to $A[i,:]$ as its $i$-th row and $A[:,j]$ as its $j$-th column and for a vector $\boldsymbol{v}$, $v[k]$ is its k-th component. For two matrices $\boldsymbol{A} \in \mathbb{R}^{m \times N_A}$, $\boldsymbol{B} \in \mathbb{R}^{m \times N_B}$, $(\boldsymbol{A}; \boldsymbol{B}) \in \mathbb{R}^{m \times (N_A + N_B)}$ refers to the concatenation along the column axis. For $C > 0$ and a vector $\boldsymbol{x} \in \mathbb{R}^C$, $\sigma(\boldsymbol{x}) \in \mathbb{R}^C$ is the softmax vector defined as $\sigma(\boldsymbol{x})[c] = \exp(x[c])/(\sum_{j=1}^{C} \exp(x[j]))$. In Section 3, we use the $r$ superscript to indicate that the model is only randomly initialized (and not trained). In Section 4, we say that a random event $E$ occurs with high probability if $\mathbb{P}[E] \geq 1 - e^{-\mathrm{poly}(d)}$.

## 2 SETTING

### 2.1 SCALED MODELS AND FEATURES

This paper aims to obtain insights into the features of overparameterized networks in comparison to the underparameterized ones given their superior performance. For this comparison, we utilize

– **Base network (High-width)**: We start with a base overparameterized (with respect to the dataset) network $M_1$ with $L$ layers and widths $n_1, \ldots, n_L$
– **Low-width networks**: We then create scaled models $M_\alpha$ with $\alpha \in [0, 1)$, which have the same architecture as $M_1$ but with scaled widths $\lfloor \alpha \cdot n_1 \rfloor, \ldots, \lfloor \alpha \cdot n_L \rfloor$. This width scaling ensures that all the models have the same inductive bias (essential good performance).

We explore how the features of the base model compare to those of low-width networks as we change $\alpha$. While the low-width networks may not always be underparameterized for each $\alpha$, this comparison provides insights into impact of the degree of parameterization as $\alpha$ decreases.

We focus on a multi-classification setting with $C$ classes, where a model is trained on the dataset $D_{\text{train}}$ and evaluated on the test dataset $D_{\text{test}}$. Given a datapoint $\boldsymbol{x}_k \in \mathbb{R}^d$, a neural network $M_\beta$ predicts the class $c \in \{1, \ldots, C\}$ with probability

$$\mathbb{P}[\hat{y}_k = c] = \sigma(M_\beta(\boldsymbol{x}_k))[c] = \sigma(\boldsymbol{W}^{(L+1)} m_\beta(\boldsymbol{x}_k))[c] = \sigma\Big( \sum_{s=1}^{\alpha n_L} \boldsymbol{W}^{(L+1)}[c, s] m_\beta(\boldsymbol{x}_k)[s] \Big), \quad (2.1)$$

where $m_\beta \colon \mathbb{R}^d \to \mathbb{R}^{\beta n_L}$ is the function computing the activation at layer $L$ (which we also name feature mapping) and $\boldsymbol{W}^{(L+1)} \in \mathbb{R}^{C \times \beta n_L}$ are the weights of the last layer. The model $M_\beta$ makes predictions based on a *linear combination of the features* $\{m_\beta(\boldsymbol{x}_k)[s]\}_{s=1}^{\beta n_L}$, which capture the information extracted from the input by the network.

**Definition 2.1** (Features). *Let $M_\beta$ be a model with $\beta \in [0, 1]$ and $\{(\boldsymbol{x}_i, y_i)\}_{i=1}^N$ be a dataset. Let $m_\beta \colon \mathbb{R}^d \to \mathbb{R}^{\beta n_L}$ be the feature mapping of $M_\beta$. Then, we define the the set of features $\mathcal{M}_\beta$ as*

$$\mathcal{M}_\beta := \Big\{ \mathscr{M}_\beta[:, 1], \ldots, \mathscr{M}_\beta[:, n_L] \Big\}, \quad (2.2)$$

*where $\mathscr{M}_\beta = [m_\beta(\boldsymbol{x}_1)^\top, \ldots, m_\beta(\boldsymbol{x}_N)^\top] \in \mathbb{R}^{N \times \beta n_L}$.*

### 2.2 FEATURE SPAN ERROR

We are interested in understanding how the features learned by overparameterized networks differ from the features learned by low-width networks. In particular, we want to answer the following questions: Do the features of overparameterized networks exhibit greater expressivity than the features of low-width networks, given that they tend to perform better? As we have seen, the model $M_\beta$ makes predictions based on a linear combination of the features $\{m_\beta(\boldsymbol{x}_k)[s]\}_{s=1}^{\alpha n_L}$. Therefore, one of the goals of this paper is to investigate whether the features learned by underparameterized networks can be span by the features learned by low-width networks.

**Definition 2.2.** *[FSE($\mathcal{M}_\beta \to \mathcal{M}_\gamma$)] Let $M_\beta$ and $M_\gamma$ be two models where $\beta, \gamma > 0$. Consider the following ridge regression problems for each $j \in \{1, \ldots, \gamma n_L\}$,*

$$\min_{\boldsymbol{c}^{(j)} \in \mathbb{R}^{\beta n_L}} \; \Big\| \sum_{k=1}^{\beta n_L} c_k^{(j)} \mathcal{M}_\beta[:, k] - \mathcal{M}_\gamma[:, j] \Big\|_2^2 + \frac{\lambda}{2} \|\boldsymbol{c}^{(j)}\|_2^2 := \mathcal{L}(\boldsymbol{c}^{(j)}), \quad \text{where } \lambda > 0. \quad \text{(Reg)}$$

*Let $(R^2)^j$ be the $R^2$-error for the regression (Reg). The feature span error FSE($\mathcal{M}_\beta \to \mathcal{M}_\gamma$) is:*

$$\text{FSE}(\mathcal{M}_\beta \to \mathcal{M}_\gamma) := \frac{1}{\gamma n_L} \sum_{j=1}^{\gamma n_L} [1 - (R^2)^{(j)}]. \quad (2.3)$$

Definition 2.2 quantifies how well the features $\mathcal{M}_\beta$ span the features $\mathcal{M}_\gamma$. To understanding the implications of a smaller FSE score, consider the following. When $\text{FSE}(\mathcal{M}_\beta \to \mathcal{M}_\gamma) = 0$, $\mathcal{M}_\beta$ can perfectly fit $\mathcal{M}_\gamma$, making $M_\beta$ is more expressive than $M_\gamma$. As such, the span of $\mathcal{M}_\beta$ can express the same prediction distribution (2.1) as $M_\gamma$. In appendix, we describe how we solve (Reg).

Using the FSE, we thus measure whether a model that uses the linear combination of its features can approximate the features of a target model. Therefore, in this paper, the expressivity of the model is measured through the linear combination of its learned features. This definition of expressivity is different from literature's classical definition for a neural network's expressivity which defines it as the set of functions a neural network can express for any choice of parameters. This classical definition does not consider the factors (such as optimization bias) which may make different width neural networks learn some distinct features.

## 2.3 CONCATENATION OF LOW-WIDTH NETWORKS

While it is believable that low-width networks features are less expressive just because they have fewer features or parameters, in this paper, we ask a more challenging question: even if we concatenate a large number of low-width networks trained independently with different initializations, so that the total number of features/parameters match that of the overparameterized network, do the concatenated low-width features have the same expressivity as the overparameterized ones?

**Definition 2.3** (Concatenation of features). *Let $M_\alpha^{(1)}, \ldots, M_\alpha^{(U)}$ be low-width networks with the same architecture that have been independently trained with different initializations. The concatenation of features $\mathcal{S}_\alpha^{(U)}$ is defined as $\mathcal{S}_\alpha^{(U)} := (\mathcal{M}_\alpha^{(1)}, \ldots, \mathcal{M}_\alpha^{(U)}) \in \mathbb{R}^{m \times \alpha n_L U}$.*

For $U$ in Definition 2.3, we consider the two following values:

- $\bar{U}$: the smallest integer such that $\mathcal{S}_\alpha^{(\bar{U})}$ has at least as many features as $\mathcal{M}_1$, i.e., $\bar{U} = \lceil 1/\alpha \rceil$.
- $U^*$: the smallest integer such that the models $M_\alpha^{(1)}, \ldots, M_\alpha^{(U)}$ collectively have at least the same number of parameters as $M_1$.

## 2.4 FEATURE PERFORMANCE

While the FSE method provides insights into the feature expressivity, it does not exactly inform us on the predictive power of the features. Therefore, we introduce the feature performance (FP) to measure the performance obtained by a set of features on a given task.

**Definition 2.4** (Feature performance). *Let $D_{train}$ and $D_{test}$ be training and test datasets of a task. Let $\mathcal{M}$ be a set of features. Assume that we train a linear classifier on top of the features $\mathcal{M}$ on $D_{train}$. Then, the feature performance $FP(\mathcal{M})$ is the accuracy of this classifier on $D_{test}$.*

A plausible hypothesis to explain the superior performance of overparameterized networks over low-width ones is: $S_\alpha^{(U)}$ cannot fully learn the features $M_1$ and this uncaptured part in the $\mathcal{M}_1$ features may be responsible for the superior performance of overparameterized networks. We refer to the part of the features that the other model cannot capture as *feature residuals*. These residuals are what enable overparameterized and low-width networks to express unique functions.

**Definition 2.5** (Feature residuals). *After solving (Reg) with regressors $\mathcal{M}_\beta$ and targets $\mathcal{M}_\gamma$, we obtain the prediction $\widehat{\mathcal{M}}_\gamma[j,:] = \sum_{k=1}^{n_L} \hat{c}_k^{(j)} \mathcal{M}_\beta[k,:]$. The feature residuals $R(\mathcal{M}_\beta \to \mathcal{M}_\gamma)$ are*

$$R(\mathcal{M}_\beta \to \mathcal{M}_\gamma) := \mathcal{M}_\gamma - \widehat{\mathcal{M}}_\gamma. \qquad (2.4)$$

In Section 3, we focus on the overparameterized feature residuals $R(\mathcal{S}_\alpha^{(U)} \to \mathcal{M}_1)$ –the features of overparameterized that low-width cannot fully capture– and on the low-width feature residuals $R(\mathcal{M}_1 \to \mathcal{S}_\alpha^{(U)})$ –the features of low-width that overparameterized cannot fully capture.

## 3 NUMERICAL EXPERIMENTS

We state our main results in this section. We apply the FSE and FP methods to image and text classification settings and state our empirical results.

## 3.1 EXPERIMENTAL SETUP

For the computer vision experiments, we trained VGG-16 (Simonyan & Zisserman, 2014) and ResNet-18 models (He et al., 2016) on CIFAR-10 (Krizhevsky et al., 2009) as our base models. Ad-

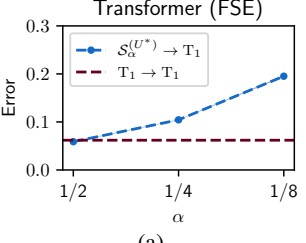 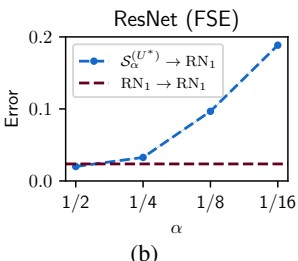 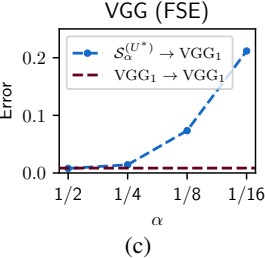

Figure 1: **FSG with respect to overparameterized features (after training).** Figures 1a, 1b and 1c display $\mathrm{FSE}(\mathcal{S}_\alpha^{(U^*)} \to \mathcal{M}_1)$ in blue line and $\mathrm{FSE}(\mathcal{M}_1 \to \mathcal{M}_1)$ in red line in the Transformer, ResNet and VGG settings. While mildly low-width networks ($\alpha = 1/2$) can fit the features of a trained overparameterized network well, low-width models ($\alpha = 1/8$ or $\alpha = 1/16$) have significantly lower performance.

ditionally, we conducted natural language processing (NLP) experiments by training a Transformer model with 4 hidden layers, hidden size 128, MLP dimension 1024, and 4 attention heads (base model) on the MNLI text classification dataset (Williams et al., 2018). The aforementioned models are overparameterized since they have more parameters than the number of training examples. The scaled models are denoted as $\mathrm{VGG}_\alpha$, $\mathrm{RN}_\alpha$, and $\mathrm{T}_\alpha$ for VGG-19, ResNet-18, and Transformer models, respectively. In this paper, we focus on small-scale settings because computing FSE is computationally expensive. Indeed, solving (Reg) involves a multi-target regression problem with a large number of regressors and targets. Moreover, we report the FSE results using the validation errors although our empirical findings are consistent with the test errors. All the results are averaged over 4 seeds. Further experimental details can be found in the Appendix.

## 3.2 CONCATENATION OF LOW-WIDTH NETWORKS CANNOT FULLY CAPTURE THE FEATURES OF OVERPAMETRIZED ONES AFTER TRAINING

$\mathrm{FSE}(\mathcal{S}_\alpha^{(U)} \to \mathcal{M}_1)$ alone does not necessarily reveal whether width-$\alpha$ networks features can capture overparameterized network features (up to a good degree). In this context, a good point of reference is $\mathrm{FSE}(\mathcal{M}_1 \to \mathcal{M}_1)$, in which a network fits another of the same type but with different initialization. Indeed, this latter indicates the minimum error (due to different randomness used in the training) that can be achieved by networks with the same structure and parameterization. Therefore, we introduce the feature span gap (FSG):

$$\mathrm{FSG}(\mathcal{S}_\alpha^{(U)} \to \mathcal{M}_1) := \mathrm{FSE}(\mathcal{S}_\alpha^{(U)} \to \mathcal{M}_1) - \mathrm{FSE}(\mathcal{M}_1 \to \mathcal{M}_1). \tag{3.1}$$

A positive FSG indicates that width-$[\alpha]$ networks cannot fully capture the overparameterized features well while a negative gap suggests the opposite. We now present our first result.

**Empirical finding 3.1.** *For $\alpha < 1/2$, concatenation of low-width network features $\mathcal{S}_\alpha^{(U^*)}$ cannot fully capture the features of overparameterized networks $\mathcal{M}_1$.*

Empirical finding 3.1 states that even when concatenating many low-width networks, $\mathcal{S}_\alpha^{(U^*)}$ does not manage to capture $\mathcal{M}_1$ as well as another overparameterized network with a different initialization would do. We choose $U = U^*$ from the two options given in Definition 2.3 to concatenate more underparameterized networks and achieve a more robust empirical finding.

We establish Empirical finding 3.1 based on Figures 1 and 7, which provide the following observations:

– **For trained networks, $\alpha$ decreases $\Rightarrow$ FSG increases.** In Figure 1, as we decrease $\alpha$, $\mathrm{FSG}(\mathcal{S}_\alpha^{(U)} \to \mathcal{M}_1)$ gets larger implying that $\mathcal{S}_\alpha^{(U)}$ gets worse at capturing $\mathcal{M}_1$.

– **For random networks, $\alpha$ decreases $\Rightarrow$ FSG decreases.** We consider a baseline experiment where we measure $\mathrm{FSG}(\mathcal{S}_\alpha^{(U),r} \to \mathcal{M}_1^r)$ where $\mathcal{S}_\alpha^{(U),r}, \mathcal{M}_1^r$ are the features at initialization. This experiment verifies where Empirical finding 3.1 is specific to the trained networks. Figure 7 shows that the random low-width features can capture the overparameterized ones. Since $\mathcal{M}_1^r, \mathcal{S}_\alpha^{(U),r}$ are sampled from the distribution, it is indeed expected that they share similar features.

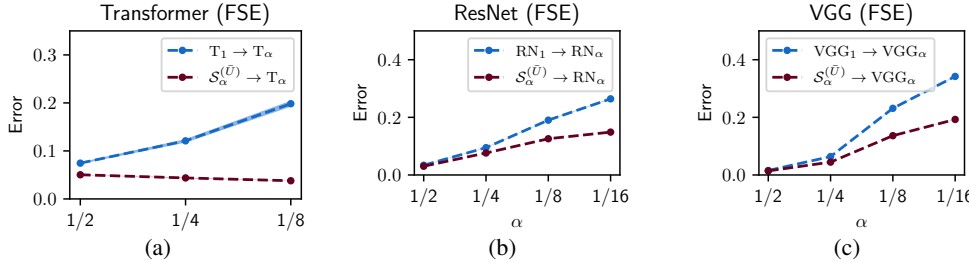

Figure 2: **FSG with respect to low-width concatenated networks features (after training).** Figures 2a, 2b and 2c display FSE($\mathcal{M}_1 \to \mathcal{S}_\alpha^{(\bar{U})}$) in blue line and FSE($\mathcal{S}_\alpha^{(\bar{U})} \to \mathcal{M}_\alpha$) in red line in the Transformer, ResNet and VGG settings. As $\alpha$ decreases, the models have lower width and $\mathcal{M}_1$ further struggles to capture $\mathcal{M}_\alpha$.

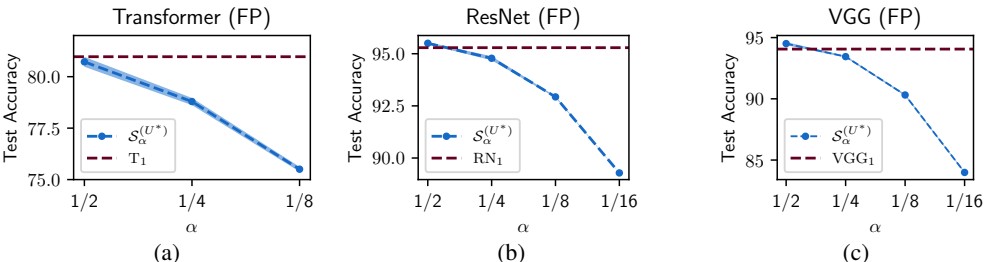

Figure 3: **Feature Performance.** Figures 3a, 3b, 3c display the feature performance obtained by concatenated low-width in blue line and by overparameterized networks in red line. As $\alpha$ decreases, low-width networks fail to match the test accuracy of a single overparameterized network.

Empirical finding 3.1 alone does not necessarily imply that overparameterized models have a superior expressivity over low-width models. Low-width models may also learn some features that overparameterized models cannot express as well.

**Overparameterized networks cannot fully capture the features of sufficiently low-width ones.**
We establish this finding based on Figures 2 and 8. To strengthen our result, we set $U = \bar{U}$ (instead of $U = U^*$) to decrease the ability of low-width networks to fit the features to get a more robust result. In Figure 2, we observe that the FSG is large for sufficiently low-width models ($\alpha = 1/8$ or $\alpha = 1/16$), while it is close to zero for mildly low-width models. In Figure 8, we observe that the FSE is almost negligible, which is consistent with the observations in Figure 7.

In summary, we have showed that the low-width networks cannot fully capture the features of overparameterized networks and the converse also holds.

### 3.3 CONCATENATIONS OF LOW-WIDTH NETWORKS CANNOT ACHIEVE THE PERFORMANCE OF A SINGLE OVERPARAMETERIZED NETWORK

Subsection 3.2 focused on the expressivity of overparameterized and low-width networks features. In this section, we investigate the performance of these two feature sets.

**Empirical finding 3.2.** *For $\alpha < 1/2$, we observe that the features of low-width networks $\mathcal{S}_\alpha^{(U^*)}$ cannot perform as well as the overparameterized features $\mathcal{M}_1$ i.e. for some small constant $\delta > 0$,*

$$\text{FP}(\mathcal{M}_1) - \text{FP}(\mathcal{S}_\alpha^{(U^*)}) \geq \delta. \tag{3.2}$$

Empirical finding 3.2 is based on Figure 3. We observe that FP($\mathcal{S}_\alpha^{(U)}$) is significantly lower than FP($\mathcal{M}_1$), and the performance gap between them increases as $\alpha$ decreases. Overall, Empirical findings 3.1 and 3.2 demonstrate that the low-width features do not capture certain parts of the features of overparameterized networks –which are the overparameterized feature residuals– that contribute to their superior performance. With that insight, we now investigate the contribution of feature residuals to the model performance.

**Impact of the feature residuals on the performance.** Empirical finding 3.2 suggests that the feature residuals $R(\mathcal{S}_\alpha^{(U)} \to \mathcal{M}_1)$ significantly improve the performance of the model contrary to the residuals $R(\mathcal{M}_1 \to \mathcal{S}_\alpha^{(U)})$. We empirically validate this hypothesis using Figures 4.

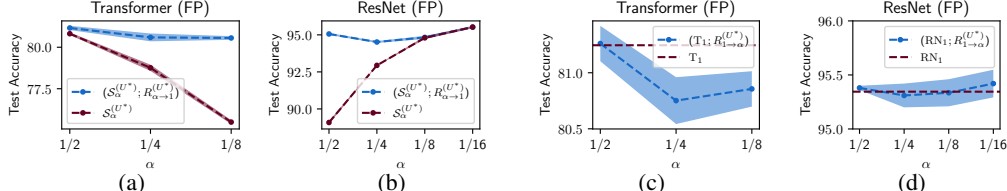

Figure 4: **Contribution of feature residuals to test accuracy.** Figures 4a and 4b compares the concatenated low-width network $\mathcal{S}_\alpha^{(U^*)}$ (red line) with the same model to which we append feature residuals $R(\mathcal{S}_\alpha^{(U^*)}) \to \mathcal{M}_1$) –shortly $R_{\alpha \to 1}^{(U^*)}$ – in blue line. These plots show that as $\alpha$ decreases, the test accuracy gains brought by the residuals increases. Figures 4c and 4d show that adding the residuals $R(\mathcal{M}_1 \to \mathcal{S}_\alpha^{(U^*)})$ –shortly $R_{1 \to \alpha}^{(U^*)}$ – does not increase or lowers the performance of $\mathcal{M}_1$ (a result of adding redundant features).

– In Figures 4a and 4b, the feature concatenation $(R(\mathcal{S}_\alpha^{(U)} \to \mathcal{M}_1); \mathcal{S}_\alpha^{(U)})$ outperforms $\mathcal{S}_\alpha^{(U)}$. Thus, $R(\mathcal{S}_\alpha^{(U)} \to \mathcal{M}_1)$ helps low-width models to span new functions that improve their performance.

– In Figures 4c and 4d, adding $R(\mathcal{M}_1 \to \mathcal{S}_\alpha^{(U)})$ to $\mathcal{M}_1$ does not improve the performance of $\mathcal{M}_1$.

## 4 HOW DO WIDE MODELS CAPTURE FEATURES THAT SHALLOW ONES CANNOT?

Our empirical results highlight a main observation: overparameterized networks can learn some essential features that low-width models cannot capture. In this section, we present a toy setting to illustrate this observation. We demonstrate that certain features can only be learned by overparameterized networks, while the concatenation of low-width networks fails to capture them effectively. Note that our setting partially covers the results from Section 3, since we do not show that overparameterized networks cannot learn some features learned by underparameterized networks. Lastly, our setting involves deep neural networks and because the analysis of their dynamics is an open question, we only validate the mechanism empirically and leave the formal proof for future work.

**Data distribution.** Let $v_1, \ldots, v_5 \in \mathbb{R}^d$ be an orthonormal set of vectors which we name *signal vectors*. Each sample consists of an input $X$ and a label $y$ such that:

1. $X = (X[1], \ldots, X[7]) \in \mathbb{R}^{7d}$, where each patch $X[j] \in \mathbb{R}^d$.
2. Sample $s_1, \ldots, s_5$ such that each $s_j$ is uniformly distributed over $\{-1, 1\}$.
3. The label $y$ is defined by $y = \mathrm{sgn}(s_1 + s_2 + s_3 \cdot s_4 \cdot s_5)$.
4. Signal patches: Uniformly sample indices $\ell_1(X), \ldots, \ell_5(X)$ such that
$$X[\ell_j(X)] = s_j \cdot v_j,$$
5. Noisy patches: $X[j] \sim \mathcal{N}(0, \xi^2(\mathbf{I}_d - \sum_{i=1}^5 v_i v_i^\top))$, for the remaining patches.

We refer to the signal vectors $v_j$ whose sign $s_j$ are involved in the addition (resp. multiplication) operator in item 3 as the addition (resp. multiplication) signal vectors. These signal vectors represent the features of the task but we do not refer to them as such to avoid a confusion with Definition 2.1. Our setting is a binary classification problem on a synthetic dataset of images, where each image is made of 7 patches. Each patch is either a random Gaussian vector or a signal vector. The highlight of our setting is that the label is a linear combination of a sum and product over the signs $s_j$. This choice leads overparameterized networks to learn some features that low-width ones cannot capture as we explain below. We choose $d = 17$ in our experiments.

**Learner model.** We train a model $M$ which is a 4-hidden layer multilayer perceptron with ReLU activation. Given an input $X$, the output of $M$ is
$$f_M(X) = g \circ \mathrm{ReLU}(W_1 X - \mathbf{1}_{7d}). \tag{MLP}$$
where $W_1$ denotes the first layer weights, and $g$ represents the subsequent three layers. (MLP) is a standard MLP except that we substract the pre-activation of the first layer by $\mathbf{1}_{P \cdot d}$, which is key to discriminate the learning of low-width and overparameterized models as explained below.

**Initialization.** The way we initialize neurons in the first layer plays a crucial role in our setting. We initialize as $\boldsymbol{W}_1^{(0)}[j, :] \sim \mathcal{N}(0, \chi^2 \mathbf{I}_d)$ where the variance $\chi^2$ is chosen such that:

– The activation of a neuron $j$ by a signal vector $\boldsymbol{v}_k$ (i.e., $\langle |\boldsymbol{W}_1^{(0)}[j, :]|, \boldsymbol{v}_k \rangle > 1$) is a rare event.

– As such, wider networks are likely to have more neurons activated by the signal vectors $\boldsymbol{v}_k$.

– While the activation of a neuron by a noisy patch is a rare event, it is still more likely to occur compared to activation by a signal vector –since the expected norm of noisy patches is slightly larger than the norm of the signal vectors.

Furthermore, all signal vectors are orthogonal to each other and to the noisy patches. This constraints the activation of a neuron $j$ in several ways. First, a neuron is likely to be activated by at most one single signal vector at initialization. Besides, because of weight decay, the norm of the neuron is constrained throughout the training. As such, it is unlikely to be simultaneously activated by two different signal vectors later on. Lastly, for similar reasons, a neuron activated by a noisy patch at initialization is unlikely to be activated by a signal later on. See the Appendix for more details.

Table 1: **Performance of concatenations of low-width and overparameterized networks** Our study focus on analyzing first layer only where networks learn signal patches, not the upper layers that learn multiplication and addition of the signs. Hence, $U^*$ is calculated according to the first layer. We refer to the largest three network as overparameterized while referring to the others as underparameterized.

| $n_1$ | $U^*$ | Average train accuracy (%) | Average test accuracy (%) |
|---|---|---|---|
| 100 | 300 | $75.01 \pm 0.07$ | $74.12 \pm 0.88$ |
| 250 | 120 | $75.12 \pm 0.25$ | $74.23 \pm 0.74$ |
| 10,000 | 3 | $100.00 \pm 0.00$ | $100.00 \pm 0.00$ |
| 15,000 | 2 | $100.00 \pm 0.00$ | $100.00 \pm 0.00$ |
| 30,000 | 1 | $100.00 \pm 0.00$ | $100.00 \pm 0.00$ |

**Training.** We train our model using binary cross entropy and normalized stochatic gradient descent for faster convergence. See the Appendix for more details.

We measure the performance of a model through their ability to learn the signal vectors. A model $M$ learns the signal vector $\boldsymbol{v}_k$ if it has at least one activated neuron $j$ which is highly correlated with $\boldsymbol{v}_k$,

$$\left\langle \frac{|\boldsymbol{W}_1^{(final)}[j, :]|}{\|\boldsymbol{W}_1^{(final)}[j, :]\|_2}, \boldsymbol{v}_k \right\rangle \geq 1 - \delta \text{ (correlation) and } \langle \boldsymbol{W}_1^{(final)}[j, :], \boldsymbol{v}_k \rangle > 1 \text{ (activated)}, \quad (4.1)$$

where $\boldsymbol{W}_1^{(final)}$ are the first layer weights after training and $\delta > 0$ is some small constant. On the other hand, $M$ does not learn $\boldsymbol{v}_k$ if $\langle \boldsymbol{W}_1^{(final)}[j, :], \boldsymbol{v}_k \rangle \leq 1$.

**Empirical finding 4.1.** *Let $M_1$ be overparameterized network and $M_\alpha^{(1)}, \ldots, M_\alpha^{(U^*)}$ be low-width networks with different initialization. After independently training these models, we observe that*

1. *$M_1$ learns all the signal vectors.*
2. *The concatenation of low-width networks learn the addition signal vectors. However, none of the networks in the concatenation learns the multiplication signal vectors.*

We now describe some important dynamics of the learning process as follows:

– **Initialization**: In order to learn $\boldsymbol{v}_k$, the network needs to have some neurons that are *activated*.

– **Neuron deactivation by weight decay:** A neuron is subjected to two forces: weight decay, which attracts the neuron to have a zero norm and leads to its deactivation and the gradient, which pushes the neuron in a direction that potentially increases its correlation with $\boldsymbol{v}_k$ (see Figure 6).

– **Irreversibility of Deactivation:** Neurons that becomes deactivated (or not activated before) are unlikely to be activated by $\boldsymbol{v}_k$ during training. Indeed, for deactivated neurons, the only term in the gradient is the one arising from weight decay, which makes the neuron shrink further.

– **Low-width networks have a few neurons activated by signal vectors at initialization**: Each small network have at most a few neurons that are activated by signal vectors (see Figure 9). This is due to their small number of neurons.

– **Overparameterized networks have many activated neurons at initialization**: because of their large width, for each signal vector $\boldsymbol{v}_k$, there exists many neurons that are activated.

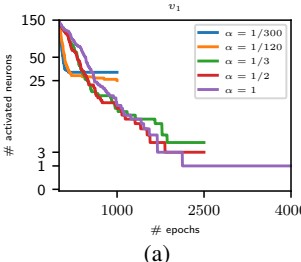 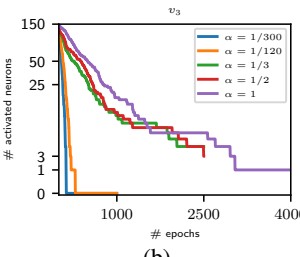 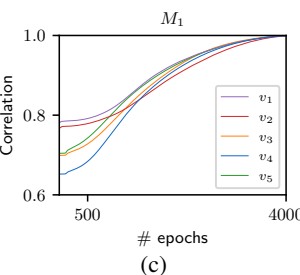

| (a) | (b) | (c) |

Figure 5: Figures 5a and 5b display the total number of activated neurons by $\boldsymbol{v}_1$ (addition signal vector) and $\boldsymbol{v}_3$ (multiplication signal vector) throughout the training process. We display these curves for overparameterized ($\alpha = 1$) and low-width concatenations ($\alpha \in \{1/300, 1/120, 1/3, 1/2\}$) networks. Both models learn the addition vectors but the low-width ones fail to learn $\boldsymbol{v}_3$ since the curves for $\alpha \in \{1/300, 1/200\}$ quickly collapse to 0. Figure 5c displays the evolution of the correlation between an overparameterized model's neurons and the signal vectors. The correlations gradually increase to 1, meaning that $M_1$ has learned the signal vectors.

During the training, overparameterized models learn all the signal vectors while the neurons of concetenations of low-width networks unlearn some signal vectors.

– **Both models learn the addition signal vectors.** Low-width and overparameterized networks keep a non-zero correlation with $\boldsymbol{v}_1$ by the end of the training (Figure 5a).

– **Low-width networks unlearn the multiplication signal vectors**: Figure 5b shows that the neurons' correlations with $\boldsymbol{v}_3$ collapse to zero for low-width networks while it stays non-zero for overparameterized ones.

**What is special about the multiplication features?** A model can independently learn the addition signal vectors $\boldsymbol{v}_1$ and $\boldsymbol{v}_2$ for an improved model accuracy. However, the model must learn the multiplication vectors $\boldsymbol{v}_3$, $\boldsymbol{v}_4$, and $\boldsymbol{v}_5$ simultaneously for improved accuracy:

– Addition case: assume that a model $M$ has learned $\boldsymbol{v}_1$. Then, it can predict its sign $s_1$, and the accuracy is $\mathbb{P}_{(\boldsymbol{X},y)}[Y = y|s_1] = 0.75$, better than random predictor.

– Multiplication case: assume a model has learned $\boldsymbol{v}_3$, $\boldsymbol{v}_4$ and not $\boldsymbol{v}_5$. Its accuracy is then $\mathbb{P}_{(\boldsymbol{X},y)}[Y = y|s_3, s_4] = 0.5$, the same as the random predictor. Learning all but one provides no value.

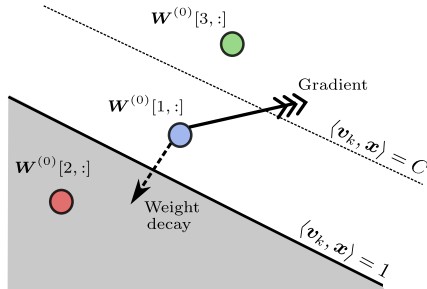

Figure 6: **Sketch of a three neurons at initialization.** The grey and white areas respectively represent the areas where neurons are activated and deactivated by the signal vector $\boldsymbol{v}_k$. $\boldsymbol{W}^{(0)}[2,:]$ is deactivated while $\boldsymbol{W}^{(0)}[3,:]$ is far from the activation hyperplane and is likely to stay activated during training. $\boldsymbol{W}^{(0)}[1,:]$ is also activated but may eventually get deactivated because of the weight decay and nosier signal from the gradient, particularly during the early stages of training with active noisy patches.

## 5 CONCLUSION AND FUTURE WORK

In this work, we study the difference between overparameterized and underparameterized networks in terms of expressivity and performance. Through experiments on CIFAR-10 and MNLI, we have demonstrated that even after concatenating many models, underparameterized features cannot cover the span nor retrieve the performance of overparameterized features. Our work suggests some future work. We consider a deep network in Section 4 since our labeling function involves a multiplication operation. It is known that these labels cannot be recovered by a 1-hidden layer network with finite width (Allen-Zhu & Li, 2020). It would be interesting to find a 1-hidden layer network setting where we can prove the benefits of overparameterized features. Lastly, our setting does not exactly match with our empirical results since we do not demonstrate that underparameterized networks learn some features that overparameterized ones cannot capture.

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

## A   ADDITIONAL DETAILS AND RESULTS ON THE FSG METHOD

In this section, we provide additional implementation details on the FSG method and present some experiments that were mentioned in the main paper. In Equation Reg, we present the method to solve the regression problem equation Reg. In Subsection A.2, we present additional details about our experiments. Lastly, in Subsection A.3, we present our FSG results on randomly initialized networks. These results were mentioned in Section 3.

### A.1   METHODOLOGY TO SOLVE THE REGRESSION PROBLEM.

The regression problem equation Reg is defined in the context of a particular task with fixed training and test datasets, which we denote by $D_{train}$ and $D_{test}$, respectively. We now explain some important details about how we implement the regression.

– **Why adding a $L_2$-regularization?**: We add a $L_2$-regularization in equation Reg for multiple reasons. First, it helps with multicollinearity and ill-conditioning. Second, it provides regularization, which leads to better out-of-sample predictions. Finally, it has a closed-form solution that makes it possible to solve the regression problem for all target variables simultaneously, which is computationally more efficient than solving the regression problem for each target variable separately. The closed-form solution requires the inversion of a matrix that can be precomputed once and used to solve the regression problem for any target variable, making it faster.

– **Cross-validation**: We independently solve equation Reg for each $j \in \{1, \ldots, \lfloor \alpha n_L \rfloor\}$ by using the RidgeCV function from the scikit-learn package (Pedregosa et al., 2011). It implements leave-one-out cross-validation (LOOV) to tune the penalty parameter $\lambda$. In our experiments we tune $\lambda$ over the interval $[1e - 5, 3e5]$.

– **Datasets in the regression**: The training (resp. test) dataset of the regression consists of the features extracted from the training (resp. test) dataset $D_{train}$ (resp. $D_{test}$) of the task.

– **Reporting FSE**: We only present the validation errors of equation Reg for FSE in our empirical findings of section 3. That is because the training dataset is much larger compared to the test dataset, and the relationship between the features of different width networks may differ between the datasets $D_{train}$ and $D_{test}$ due to the weights of the trained models (for that matter the features of the networks) being dependent on $D_{train}$ but not $D_{test}$. We discuss this issue further in Subsection D.4, where we present regression results using a training dataset consisting of data points from both $D_{train}$ and $D_{test}$ (training and test datasets of the task). Nevertheless, the empirical findings remain consistent for the test errors (Subsection D.3), although there may be a gap between the test and validation errors in some cases.

### A.2   ADDITIONAL DETAILS ABOUT EXPERIMENTAL SETUP

Regarding the training procedure of the vision experiments, all the models are trained with standard augmentations techniques and using SGD with momentum 0.9, a tuned weight-decay, a tuned learning-rate and a linear decay step-size scheduler (applied by the factor of 0.1 at epochs 180 and 255). We run all the experiments for 300 epochs, with batch size 128 otherwise. For NLP, we run all the experiments for 30 epochs, using AdamW with a tuned weight-decay, a tuned learning-rate, and a cosine learning rate scheduler. Table 2 displays the average training and test accuracies of the networks used in the experiments.

We note that for each different model, we tune the learning-rate and weight-decay for a single seed and use the same hyperparameters for all other models of the same kind but with a different initialization. In our computer vision experiments, our range of learning rates is $\{0.02, 0.04, 0.06, 0.08, 0.1\}$ and for weight decay, the range is $\{7e-4, 5e-4, 9e-4, 3e-4, 5e-4, 5e-4\}$ and use dropout=0.1. In our NLP experiments, our range of learning rates is $\{5e-6, 1e-5, 5e-5, 9e-5, 1e-4, 5e-4, 9e-4, 5e-3\}$ and for weight decay, $\{1e-6, 1e-5, 5e-5, 1e-4, 5e-4, 1e-3, 1e-2, 5e-3, 5e-2\}$.

To create low-width networks from ResNet-18 and VGG-16 vision models, we reduce the number of filters or channels in each layer compared to the original architecture. Specifically, we achieve the scaled model $M_\alpha$ by adjusting the number of filters in each layer, multiplying the original number of filters by a factor of $\alpha < 1$. Similarly, for the transformer model, the hidden size is modified by scaling it down by a factor of $\alpha < 1$. This process results in the low-width model $M_\alpha$.

**Remark about the NLP experiments.** The primary focus of this paper is to analyze the impact of overparameterization on network features, driven by their superior performance. As a result, we specifically investigate scenarios where overparameterized networks exhibit significantly better performance than low-width networks. In our NLP experiments, we observed that when using a transformer model, there was no notable difference in performance between low-width and overparameterized networks. We attribute this lack of distinction to the embedding layer, which has a vocabulary size of approximately 50k, causing low-width networks to have an excessive number of parameters. To address this, we removed the embedding layer from the models studied in this paper. Additionally, we leveraged a pre-trained network to process the data and provide its hidden representation as input to these models.

## A.3 MEASURING THE FSG FOR RANDOM NETWORKS

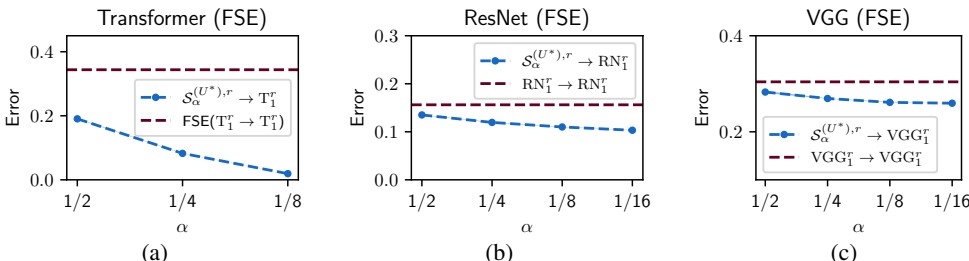

Figure 7: **FSG with respect to overparameterized features (random networks).** FSE($\mathcal{S}_\alpha^{(U^*),r} \to \mathcal{M}_1^r$) and FSE($\mathcal{M}_1^r \to \mathcal{M}_1^r$) are displayed by blue and red lines, respectively, in the settings of Transformer, ResNet, and VGG. Unlike trained networks, the concatenated random low-width network features are capable of fitting overparameterized network features effectively, regardless of their size. In fact, we observe that the fitting improves as the value of $\alpha$ decreases.

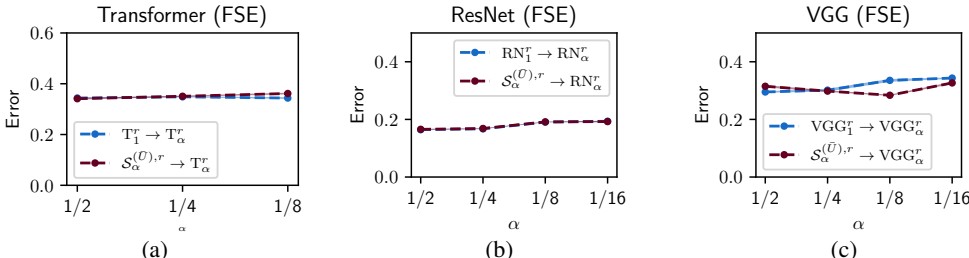

Figure 8: **FSG with respect to low-width concatenated networks features (random networks).** FSE($\mathcal{S}_\alpha^{(U^*),r} \to \mathcal{M}_1^r$) and FSE($\mathcal{M}_1^r \to \mathcal{M}_1^r$) are displayed by blue and red lines, respectively, in the settings of Transformer, ResNet, and VGG. Unlike trained networks, the concatenated random low-width network features exhibits no difference compared to overparameterized networks in capturing overparamaeterized network features.

## B ADDITIONAL DETAILS ABOUT FEATURE PERFORMANCE

In the feature performance experiments in Subsection 3.3, the models are trained with standard augmentations techniques (only for the vision experiments) and using SGD with momentum 0.9, a tuned weight-decay, a tuned learning-rate and a linear decay step-size scheduler. We run all the experiments with 40 or 80 epochs and ensure that at the end of training, the training loss is constant in all the cases.

## C ADDITIONAL DETAILS FOR OUR THEORETICAL SETTING

In this section, we provide some technical information about our initialization scheme. We then display the number of neurons activated by each feature at initialization (Figure 9).

| | | | RN$_\alpha$ | | | VGG$_\alpha$ | | | T$_\alpha$ | |
| | | | Average training | Average test | | Average training | Average test | | Average training | Average test |
| $\alpha$ | $\bar{U}$ | $U^*$ | accuracy (%) | accuracy (%) | $U^*$ | accuracy (%) | accuracy (%) | $U^*$ | accuracy (%) | accuracy (%) |
|---|---|---|---|---|---|---|---|---|---|---|
| 1 | - | - | $99.99 \pm 0.01$ | $95.29 \pm 0.14$ | - | $99.98 \pm 0.01$ | $94.00 \pm 0.01$ | - | $86.49 \pm 0.16$ | $81.06 \pm 0.18$ |
| 1/2 | 2 | 4 | $99.98 \pm 0.01$ | $94.56 \pm 0.13$ | 4 | $99.80 \pm 0.02$ | $92.82 \pm 0.13$ | 4 | $83.64 \pm 0.20$ | $79.72 \pm 0.36$ |
| 1/4 | 4 | 16 | $99.81 \pm 0.02$ | $92.72 \pm 0.25$ | 16 | $98.11 \pm 0.06$ | $90.27 \pm 0.25$ | 15 | $79.66 \pm 0.29$ | $77.63 \pm 0.21$ |
| 1/8 | 8 | 64 | $96.12 \pm 0.10$ | $89.30 \pm 0.28$ | 64 | $90.12 \pm 0.14$ | $85.90 \pm 0.21$ | 59 | $74.93 \pm 0.25$ | $73.77 \pm 0.21$ |
| 1/16 | 16 | 254 | $86.81 \pm 0.19$ | $84.30 \pm 0.28$ | 251 | $75.66 \pm 0.52$ | $76.47 \pm 0.63$ | - | - | - |

Table 2: **Average training and test accuracies of trained models**: $U^*$ is shown with respect to the largest model ($\alpha = 1$). Accuracies are rounded to the nearest second digit after decimal.

## C.1 INITIALIZATION SCHEME

**Initialization.** We assume that the first layer is initialized for all $j \in [n_1]$ as $\boldsymbol{W}_1^{(0)}[j,:] \sim \mathcal{N}(0, \chi^2 \mathbf{I}_d)$ where $\chi = d^{-(5/12)}$. The remaining parameters may be initialized as in standard ReLU neural networks, e.g., by using Kaiming initialization (He et al., 2015).

**Justification of the parameters choice.** We here justify the multiple choices made in the setting described above:

- Variances $\xi^2, \chi^2$ and activation in in the first layer: these choices are made for two reasons. First, with high probability, the norm of the noisy patches are slightly bigger than the norm of the signal ones. This incentivizes the model to have more signal-activated neurons than noise-activated neurons. That, in turn, makes it possible for some (or possibly all) signal-activated neurons to get de-activated especially early in the training when noise patches are more dominant in the objective function.

  Besides, under this parameters choice, neurons rarely activate a signal patch, i.e., for $j \in [n_1]$ and $p \in \{\ell_1(\boldsymbol{X}), \dots, \ell_k(\boldsymbol{X})\}$, the event "$\sigma(\boldsymbol{W}_1^{(0)}[j,:]^\top \boldsymbol{X}[p] - 1) > 0$" is rare. Indeed, by concentration inequality, we have

$$\mathbb{P}[\boldsymbol{W}_1^{(0)}[j,:]^\top \boldsymbol{v}_p > \chi\sqrt{d} \approx 1] \leq e^{-\mathrm{poly}(d)}. \tag{C.1}$$

- Orthogonality of features and noisy patches: given the orthogonal structure in $\mathcal{D}$, a neuron $\boldsymbol{W}_1^{(0)}[j,:]$ can only activate a single feature. Indeed, by Pythagoras theorem, we have

$$|\boldsymbol{W}_1^{(0)}[j,:]\|_2^2 \geq \sum_{p=1}^k \langle \boldsymbol{W}_1^{(0)}[j,:], \boldsymbol{v}_p \rangle^2 \tag{C.2}$$

  With high probability, we have $|\boldsymbol{W}_1^{(0)}[j,:]\|_2^2 = \chi^2 d = d^{0.02} \in [1, 2)$ and by definition, a neuron activates a feature if $\langle \boldsymbol{W}_1^{(0)}[j,:], \boldsymbol{v}_p \rangle^2 > 1$. Thus, at initialization, a neuron can only activate a single feature.

- Adding weight decay to the loss: weight decay usually leads to a solution with minimal $L_2$-norm. In our case, we show that if a neuron isn't activated by any feature or noise in the beginning, then its weight will keep shrinking and never get activated. This follows from the fact that in this case the neuron's weights is updated only by weight decay since gradient from the loss function is zero.

## C.2 ADDITIONAL IMPLEMENTATION DETAILS

In our experiments in Section 4, we set the learning rate to 0.002 and weight decay to 0.003. Depending on the value of $\alpha$, we train the model for a different time. For instance, for $\alpha \in \{1/300, 1/120\}$, we train the models for 1000 epochs, $\alpha \in \{1/3, 1/2\}$, we train for 2500 epochs and for $\alpha = 1$, 4000 epochs. This explains why the lines are discontinuous in Figure 5. In every case, we ensure that the training has ended by checking that the training loss is constant.

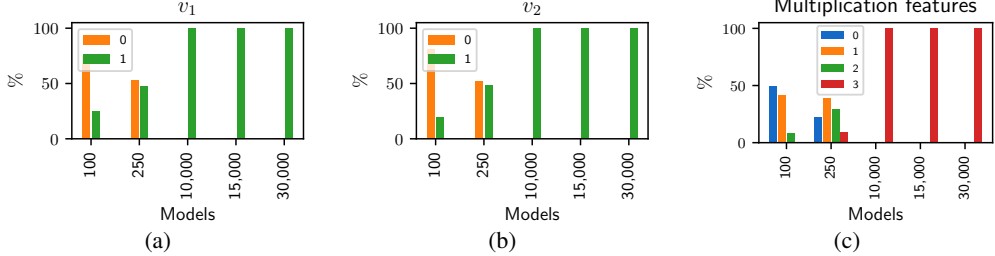

Figure 9: Barplots 9a and 9b show the percentage of networks in which the first and second addition features are activated at initialization, respectively, with 1 representing the percentage of networks with the corresponding features activated. Barplot 9c categorizes the models by the number of multiplication features that are activated at initialization.

# D    ADDITIONAL EXPERIMENTAL RESULTS

In this section, we present some empirical results that were briefly mentioned in the main paper. In Subsection D.1, we measure the performance of the feature residuals. In Subsection D.2, we append feature residuals to overparameterized and low-width networks and measure their contribution to the task performance. While in the main paper we report the validation errors for the FSG method, we report in Subsection D.3 the test errors. Lastly, in Subsection D.4, we present FSE results where the features in the regression problem are obtained from a dataset that is a mix of training and test sets.

## D.1    PERFORMANCE OF FEATURE RESIDUALS ALONE

| Features | Transformer | ResNet | VGG |
|---|---|---|---|
| $FP(R(\mathcal{M}_1 \to \mathcal{S}_\alpha^{(U^*)}))$ | $31.47 \pm 0.52$ | $17.79 \pm 0.73$ | $40.05 \pm 1.16$ |
| $FP(\mathcal{S}_\alpha^{(U^*),r})$ | $46.98 \pm 0.55$ | $34.30 \pm 0.04$ | $47.48 \pm 0.01$ |
| $FP(R(\mathcal{S}_\alpha^{(U^*)} \to \mathcal{M}_1))$ | $50.97 \pm 0.18$ | $53.46 \pm 0.03$ | $56.42 \pm 0.06$ |
| $FP(\mathcal{M}_1^r)$ | $51.37 \pm 0.28$ | $28.73 \pm 0.36$ | $19.03 \pm 1.03$ |

Table 3: **The performance (test accuracy) of residuals alone**: Lowest $\alpha$ is picked for each model in this experiment (1/16 for ResNet and VGG, and 1/8 for Transformer).

In Table 3, we present the performance of feature residuals together with the feature performance of the random networks which serve as a baseline. In this experiment, only the feature residual with the lowest $\alpha$ is used for each model (1/16 for ResNet and VGG, and 1/8 for Transformer). We observe the the residuals of the overparameterized have better or similar performance compared to the performance of its baseline (random overparameterized network features). By contrast, the residual of the low-width networks underperform its baseline (concatenation of random low-width network features).

## D.2    CONTRIBUTION OF FEATURE RESIDUALS FOR VGG

This section shows the contribution of residuals for VGG.

## D.3    TEST ERRORS FOR THE FSG METHOD

Figures 11, 12, 13, and 14 display the test errors corresponding to the figures in Sections 3 and A.3. We observe that the test errors exhibit a similar trend to the validation errors, indicating consistent empirical findings. However, there is a noticeable gap between test and validation errors in the trained networks of vision experiments. We delve into the details of this disparity in Subsection D.4 to provide a more comprehensive analysis.

Furthermore, we noticed a significant disparity between the test and validation errors in the random VGG experiment, as shown in Figures 7c and 12c. Upon closer examination, we discovered that

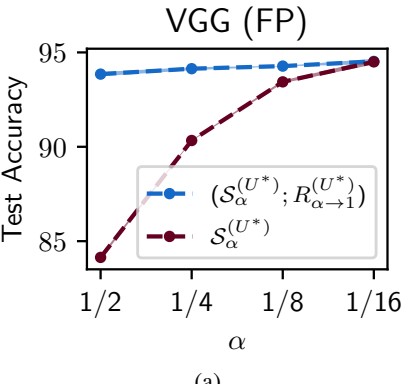
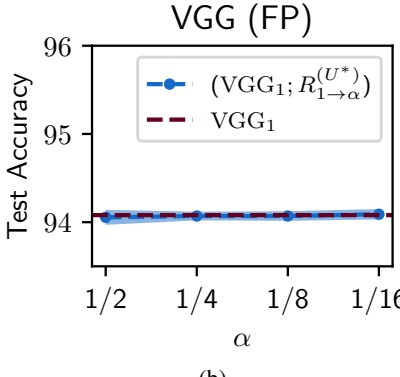

(a)                                          (b)

Figure 10: **Contribution of residuals to the test accuracy.** Figure 10a compares the concatenated low-width network $\mathcal{S}_\alpha^{(U^*)}$ (red line) with the same model to which we append feature residuals $R(\mathcal{S}_\alpha^{(U^*)} \to \mathcal{M}_1)$ –shortly $R_{\alpha \to 1}^{(U^*)}$ – in blue line. This plots show that as $\alpha$ decreases, the test accuracy gains brought by the residuals increases. Figure 10b show that adding the residuals $R(\mathcal{M}_1 \to \mathcal{S}_\alpha^{(U^*)})$ –shortly $R_{1 \to \alpha}^{(U^*)}$ – does not increase the performance of $\mathcal{M}_1$

certain features of the overparameterized network led to test errors that were notably larger than one. To improve the statistical accuracy of our results, we decided to exclude the features that resulted in a test error greater than 5. This amounted to only 12 out of the total 5120 features. Figure 12d illustrates the impact of removing these outliers on the test errors. We observe that the errors depicted in Figure 12d align with those shown in Figure 7c. As a result, by removing the outliers, we achieve consistent validation and test errors. We do not report on the change in the validation error following the removal of outliers, as the change is deemed insignificant.

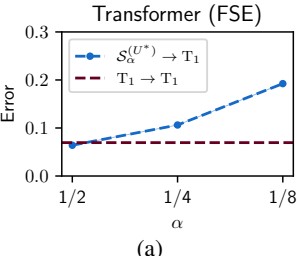
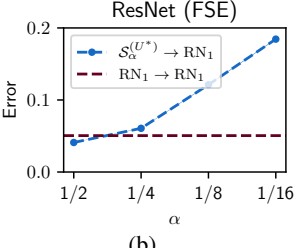
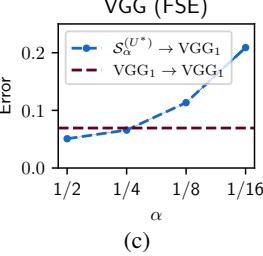

(a)                              (b)                              (c)

Figure 11: **Test Error: FSG with respect to overparameterized features (after training).** This figure displays the test errors corresponding to Figure 1.

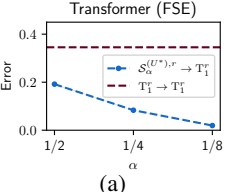
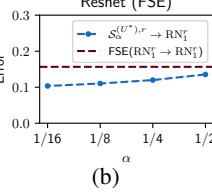
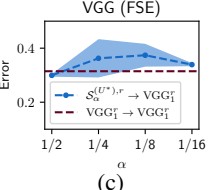
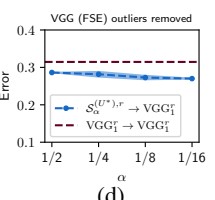

(a)                         (b)                         (c)                         (d)

Figure 12: **Test Error: FSG with respect to overparameterized features (random networks).** Figures 12a and 12b display the test errors of Transformer and ResNet corresponding to Figure 7. Figures 12c and 12d display the test errors of VGG corresponding to Figure 7c.

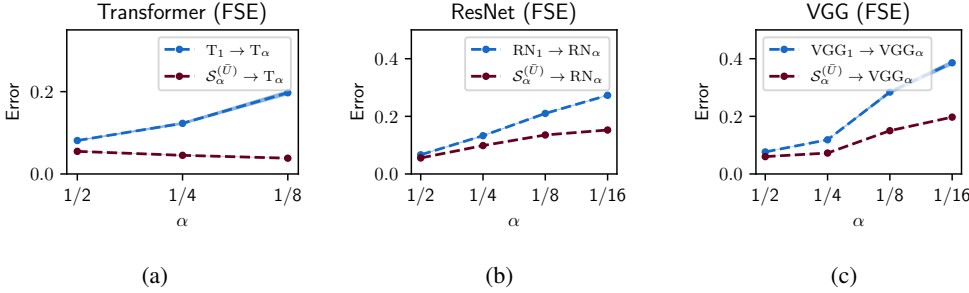

(a)            (b)            (c)

Figure 13: **Test Error: FSG with respect to low-width concatenated networks features (after training).** This figure displays the test errors corresponding to Figure 2.

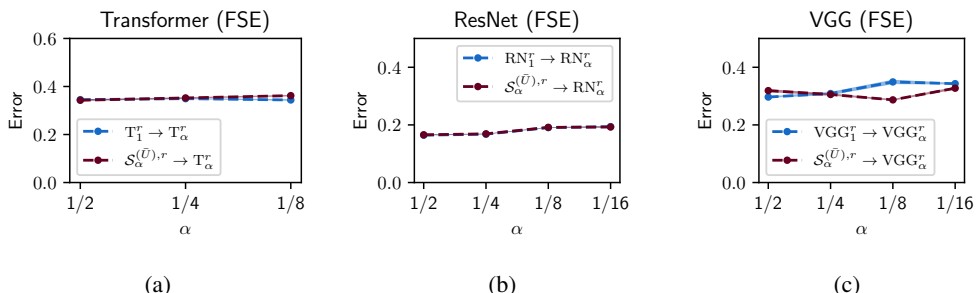

(a)            (b)            (c)

Figure 14: **Test Error: FSG with respect to low-width concatenated networks features (random networks).** This figure displays the test errors corresponding to Figure 8.

## D.4 SHUFFLED REGRESSION FOR VISION EXPERIMENTS

As shown in Subsection D.3, a non-negligible difference between validation and test errors is observed in vision experiments for the trained networks. One explanation behind this gap could be that training (resp. test) data used in training (resp. testing) the networks is the same as the training (resp. test) data used in regression. Since weights of the trained networks, for that matter features of the trained networks, is a function of training data, we may expect smaller errors with training data than with test data for FSE. To further understand this effect, we run another regression, which we call shuffled regression for vision experiments, where training data is created by randomly sub-sampling from training and test data (85%) from each) used in network training.

After comparing the regular and shuffled regression results presented in tables 4 and 5, we can observe a significant reduction in the gap between validation and test errors for the shuffled regression. This observation is in line with our earlier intuition that the dependence of trained network weights or features on the training data can lead to smaller errors on training data than on test data for FSE. Thus, we can conclude that the FSE results are sensitive to the data type (training vs test data of the task) used in the regression for vision experiments, unlike NLP experiments where a much larger training dataset is available.

| Regression | Error | FSE($RN_1 \rightarrow RN_1$) | FSE($\mathcal{S}_{1/2}^{(U^*)} \rightarrow RN_1$) | FSE($\mathcal{S}_{1/4}^{(U^*)} \rightarrow RN_1$) | FSE($\mathcal{S}_{1/8}^{(U^*)} \rightarrow RN_1$) | FSE($\mathcal{S}_{1/16}^{(U^*)} \rightarrow RN_1$) |
|---|---|---|---|---|---|---|
| Regular | valid. | 0.0236±0.000106 | 0.0201±3.1e − 05 | 0.0327±0.000165 | 0.0967±0.000294 | 0.1885±0.000111 |
|  | test | 0.0505±0.000747 | 0.041±0.000109 | 0.0607±0.00023 | 0.1212±0.000401 | 0.1845±0.000396 |
| Shuffled | valid. | 0.0279±0.00019 | 0.0234±2.9e − 05 | 0.0372±0.000178 | 0.1004±0.000264 | 0.1871±7e − 05 |
|  | test | 0.0281±0.000259 | 0.0236±3.7e − 05 | 0.0376±0.000155 | 0.102±0.000384 | 0.1908±0.000731 |

Table 4: **Shuffled Regression for Resnet**

| Regression | Error | FSE(VGG$_1 \rightarrow$ VGG$_1$) | FSE($\mathcal{S}_{1/2}^{(U^*)} \rightarrow$ VGG$_1$) | FSE($\mathcal{S}_{1/4}^{(U^*)} \rightarrow$ VGG$_1$) | FSE($\mathcal{S}_{1/8}^{(U^*)} \rightarrow$ VGG$_1$) | FSE($\mathcal{S}_{1/16}^{(U^*)} \rightarrow$ VGG$_1$) |
|---|---|---|---|---|---|---|
| Regular | valid. | $0.0084\pm6.9e-05$ | $0.0079\pm3.3e-05$ | $0.0143\pm0.000119$ | $0.0734\pm0.000126$ | $0.2116\pm0.000362$ |
| | test | $0.0694\pm0.000778$ | $0.0508\pm0.000267$ | $0.0658\pm0.000426$ | $0.1134\pm0.000214$ | $0.209\pm0.003072$ |
| Shuffled | valid. | $0.0168\pm0.000129$ | $0.0145\pm5.5e-05$ | $0.0224\pm0.000183$ | $0.0791\pm0.000118$ | $0.2101\pm0.000295$ |
| | test | $0.017\pm0.000168$ | $0.0148\pm5.9e-05$ | $0.0236\pm0.000177$ | $0.0833\pm0.000514$ | $0.2124\pm0.000795$ |

Table 5: **Shuffled Regression for VGG**

