# OpenReview forum: "How does overparametrization affect features?"
_ICLR.cc/2024/Conference — Submitted to ICLR 2024_

### Official Review · Reviewer_Y4Do · 2023-10-27

**Soundness:** 2 fair
**Presentation:** 1 poor
**Contribution:** 2 fair
**Rating:** 3
**Confidence:** 3

**Summary:**

This paper studies the influence on the number of parameters (over- vs. underparameterization) on the learned features. In particular, the work investigates whether a concatentation of independently trained underparameterized networks (with similar parameter count) retrieves the expressive power and performance of an overparameterized network. To scale networks, the work employs scaling of network widths, while keeping other network hyperparameters, such as depth, fixed (Sec. 2.1 & 2.3). To analyze feature similarity, they introduce some metrics: feature span error (Sec. 2.2), feature span gap (based on FSE; Sec. 3.2), and feature performance (Sec. 2.4). They find that underparameterized networks cannot fully capture the features of overparameterized networks, and vice versa (Sec. 3.2). Thus, they conclude that the networks seem to learn distinct features. Further, the features from the overparameterized have higher predictive prowess (Sec. 3.3). Finally, the work provides a toy example to show that some features can only be learned by overparameterized networks (Sec. 4).

**Strengths:**

* All metrics are intuitive and sound.

* The analyses are interesting.

* The toy example is interesting and sound.

* Code is provided in the supplementary results.

**Weaknesses:**

* The research question has significant flaws. That is, independently trained underparameterized networks are likely to converge to similar feature representations, as each of them tries to minimize the target loss and, thus, only the features with the largest effect on the target loss are learned (given the more restrictive capacity constraint from the width scaling). On the other hand, overparameterized networks do not suffer from such an issue and can use their larger capacity during training to learn more (and other) features to further reduce the target loss. Consequently, this raises substantial concerns about the empirical findings.

* The paper is hard to follow. For example, the varying notation makes it hard to read without keeping track of notation and resolving ambiguities. E.g., why is $\beta$ needed if $\alpha$ suffices and seems to also be used interchangeably by the authors, e.g.:
   * Eq. 2.1
   * vs. “linear combination of the features $\lbrace m_{\beta}(x_k)[s]\rbrace^{\beta n_L}_{s=1}$” (p. 3)
   * vs. $\lbrace m_{\beta}(x_k)[s]\rbrace^{\alpha n_L}_{s=1}$ (p. 4))?

* Besides the above, the paper seems partially unordered. E.g., why are the proposed metrics interleaved with the setup on how the networks are scaled?

* The introduced metric “feature performance” is only a linear probe and not an original contribution of the work.

* The feature residual analysis has contradictory results (Fig. 4a vs. 4b and Fig. 10). For the transformer setting, it is quite clear that the residual features help in predictive performance. However, for the ResNet setting this is not clear, as for $\alpha=1/8$ and $\alpha=1/16$ the difference is negligible. There is no discussion on this.

* Sec. 4 seems to reiterate the lottery ticket hypothesis (the initial weights are particularly effective or ineffective for training). It is unclear how this relates to the empirical findings of the present work.

**Questions:**

* Do the authors ensure the same random initialization between the overparameterized and the underparameterized networks (assuming that layer widths are integer multiples of $\alpha$)?

* How do the underparameterized networks perform on the target task compared to the overparameterized network?

* Seemingly, the underparameterized CIFAR-10 models improve their predictive performance for $\alpha=1/2$ in Fig. 3. Is there any explanation by the authors for why?

* How are the feature residual experiments conducted? I.e., how are these features “appended”? Is a new linear mapping $W^{(L+1)}$ learned for the additional features?

* How is the MLP scaled in Sec. 4?

## Suggestions

* As mentioned above, the paper would be easier to follow by substantially improving the presentation. For example, instead of $\beta$ and $\gamma$ in Def. 2.2, it would be easier if the authors would just use $\alpha_1$ and $\alpha_2$ instead.

* Table 1 should be within the page size limits.

---

> ### Author Response · Authors · 2023-11-22
>
> We thank Reviewer Y4Do for their comments. We address their concerns as follows:
>
> **"Independently trained underparameterized networks are likely to converge to similar feature representationss, as each of them tries to minimize the target loss "**\
> Although the reviewer's claim may have a point, it does not necessarily hold. The learning process of each network can be different depending on their initialization. It is a noisy process and smaller networks may be learning some bad quality (noisy or redundant) features and these features may not help in the test set.
> \
> Moreover, for instance, in our toy example (Section 4), all the networks have the capacity (enough neurons) to learn all the features but still they need to be large enough to learn all.
>
> **"The introduced metric “feature performance” is only a linear probe and not an original contribution of the work."**\
> We apologize for the confusion. We already state FSG is a linear probe in the paper and we will make it clearer. Our work differs from the previous work in how we utilize FSG.
>
> **"The feature residual analysis has contradictory results (Fig. 4a vs. 4b and Fig. 10)."**\
> We thank the reviewer for pointing this out. We apologize that Fig 4b is wrongly plotted. Plots in the figure are shown to be plotted for decreasing $\alpha$ but they are indeed plotted for increasing $\alpha$. That is why Figure 4b seems contradictory but the results in reality are not.
>
> **"Sec. 4 seems to reiterate the lottery ticket hypothesis (the initial weights are particularly effective or ineffective for training). It is unclear how this relates to the empirical findings of the present work."**\
> In section 4, our setup leads to many neurons to be dead at initialization. We wanted to point out that that is realistic as the lottery ticket hypothesis show.

---

> > ### Comment · Reviewer_Y4Do · 2023-11-23
> > **Re: Official Comment by Authors**
> >
> > I thank the authors for their reply. Upon reading other reviews, authors' responses, and ensuing discussions, I tend to keep my score unchanged.
> >
> > My biggest concern that independently trained underparameterized networks yield similar feature representation still remains and is at least partially acknowledged by the authors. Initialization may lead to slight differences but I would still expect that they remain quite similar to each other. Regarding Section 4, I do not see the novel insight beyond the lottery ticket hypothesis. Thus, to reiterate my question: how does this relate to the empirical findings of the work? How does it extend our understanding on feature representation learning beyond the idea of the lottery ticket hypothesis?

---

### Official Review · Reviewer_Si1X · 2023-10-29

**Soundness:** 1 poor
**Presentation:** 2 fair
**Contribution:** 2 fair
**Rating:** 3
**Confidence:** 4

**Summary:**

This work tries to investigate the difference in learned features between overparametrized and underparametrized networks. The authors explore this point by comparing regular networks (e.g. VGG-16, ResNet18) with corresponding thinner networks (e.g. ResNet18 with half channels in each layer). By using a feature cross-prediction (linear) method, the authors show the feature difference between regular networks and thinner networks. Then this work further compares the feature difference between regular networks and the concatenation of many narrower networks. Finally, the authors conclude these investigations as "overparametrized network learns more expressive than the underparameterized one".

**Strengths:**

- writing is clear and easy-to-understand.
- the idea of investigating the feature difference between over-parameterized and under-parameterized networks is interesting.

**Weaknesses:**

- The most basic requirement to verify this paper's point,  "Do the features of overparameterized networks exhibit greater expressivity than the features of low-width networks, given that they tend to perform better?", is to have **a close training performance of overparameterized network and low-width networks**. So that both networks are well-learned. Otherwise, the feature difference can come from well-learned / poorly-learned networks instead of overparameterized / underparameterized networks.
- Table 1 and Table 2 tell me the feature difference actually comes from well-learned / poorly-learned networks.  Table 1 (b) shows the FSE feature difference starts to increase at $\alpha=1/8$. meanwhile, table 2 shows the training accuracy starts to decrease at the same time ($\alpha=1/8$). Please note that when $\alpha < 1/8$,  Table 2 shows a very similar training accuracy (99.81 - 99.99) but different validation accuracy (92.72 -95.29). Table 1 ($\alpha < 1/8$) doesn't reflect feature differences.
- The proposed FSE score (Definition 2.2.) is a common metric. [1 (iclr)] shows (almost) the same feature score. [3] computes a linear regression between two sets of features. Canonical Correlation Analysis [2] also shares a close idea.

- The feature concatenation of independently learned networks was tested in [3 (icml]. But they get a very different conclusion about feature concatenation. Probably because they allow models to be well-learned. So that they avoid the well-learned / poorly-learned network problem.



[1 (lclr)] Zhang, J., & Bottou, L.  Learning useful representations for shifting tasks and distributions. https://openreview.net/pdf?id=rRgLJ8TwXe

[2] Andrew, G., Arora, R., Bilmes, J. &amp; Livescu, K.. (2013). Deep Canonical Correlation Analysis. Proceedings of the 30th International Conference on Machine Learning

[3] Kornblith, S., Norouzi, M., Lee, H., & Hinton, G. (2019, May). Similarity of neural network representations revisited. In International conference on machine learning (pp. 3519-3529). PMLR.

[4 (icml)] Zhang, J., & Bottou, L.  Learning useful representations for shifting tasks and distributions. In International Conference on Machine Learning (pp. 40830-40850). PMLR.

**Questions:**

- I suggest the author choose a regular network (e.g. resnet18) as a low-width network and use a much wider (more channels) version as the base network (overparameterized). So that you can avoid the well-learned /poorly-learned network problem.
- It is not called "shallow" in Section 4 title " HOW DO WIDE MODELS CAPTURE FEATURES THAT SHALLOW ONES CANNOT?". In general, "shallow" indicates less layers. I suggest "thin".

---

> ### Author Response · Authors · 2023-11-22
>
> We thank Reviewer Si1X for their comments. We address their concerns as follows:
>
> **"The feature difference can come from well-learned / poorly-learned networks instead of overparameterized / underparameterized networks."** \
> Our purpose is to create smaller (underparameterized) networks that has number of parameters comparable to the data size. Hence, we cannot use resnet18 as the lowest degree model in our experiments since it has much more paramaters than the data size.
> \
> Moreover, we train all the networks for 300 epochs. Since smaller networks ($\alpha \le 1/8$) do not achieve a training accuracy close to 100 with 300 epochs, we also tried training these networks much longer. Although training much longer helped somewhat increasing the training accuracy, it either decreased or very marginally increased the validation accuracy. It is also natural that smaller (underparameterized) networks may not achieve a perfect training accuracy due to their capacity.
> \
> Lastly, we concatenate many smaller networks to match the parameters of the large network. It could have been enough to match the number of features in the large network for a fair comparison. In that sense, we over-concatenate for a stronger empirical finding and still FSE scores of these concatenations are significantly high.
>
> **"The proposed FSE score (Definition 2.2.) is a common metric."**\
> We do not claim that FSE is a completely different metric from the previous ones. FSE is indeed used in [3] and we mentioned this (citing it) in our paper. The novelty of our paper lies in the use of FSE and FSG to compare the features between low- and high-width networks. Our papers also mentions some other similarity metrics too. Our work significantly differs from the literature in the way it utilizes FSE and FSG.

---

> > ### Comment · Reviewer_Si1X · 2023-11-23
> >
> > Thanks for your answer. As the title says "How does overparametrization affect features?", this work tries to investigate the features learned by overparametrized networks and compare these features with non-overparameterized (aka, underparameterized) networks. To compare features of two models (over- / under-parameterized), one requirement is to at least have a similar training loss. Otherwise, one can not know whether the different features come from over- / under-parameterized model or the well / poorly-training process.
> >
> > For example, with the exact same network (e.g. resnet18), one can train it with different hyperparameters to achieve different training losses. Then the resulting features could also be very different.

---

> > > ### Author Response · Authors · 2023-11-23
> > >
> > > We tune all the models using a variety of different hyperparameters and do not bias training process to favor larger size models.  We provide best possible training setting for each model to get their best predictive performance. We compare different size models with their best possible performances. So, they are comparable.\
> > > As we stated before, it is expected that underparameterized smaller models may not reach 100 training accuracy since they have lower capacity. This doesn't mean that they are poorly trained.

---

### Official Review · Reviewer_6Vs9 · 2023-10-30

**Soundness:** 2 fair
**Presentation:** 2 fair
**Contribution:** 2 fair
**Rating:** 3
**Confidence:** 4

**Summary:**

The paper studies the overparametrization of neural networks from the perspective of their expressive power. Specifically, the paper compares a wide network with an ensemble of shallow network that has the same width of the wide network. The paper uses a ridge regression between features to measure their expressive power. The paper demonstrates that even after concatenating many models, underparameterized features cannot cover the span nor retrieve the performance of overparameterized features. At last, the paper uses one specific case to show the difference of small and large network and what leads to the difference.

**Strengths:**

1. The paper studies an important problem of overparametrization, and show that ensemble of small models cannot recover the expressive power of overparameterized models.

2. The paper proposes FSE, which arises from ridge regression, to measure the  expressive power.

**Weaknesses:**

1. The paper does not justify why the ridge regression is an appropriate method to measure the expressive power. As it is known to all, the network is a very complicated non-linear models. The true expressive power should be analyzed in terms of the function classes of these two kinds of networks.

2. The paper does not justify why comparing overparameterized models with an ensemble of shallow models is important or meaningful. As an ensemble of small networks has fewer parameters than the large network, why is this a fair comparison?

3. The paper only provides empirical observations and lacks of theoretical analysis.

4. The mathematical symbols of the paper is a little bits complicated, which makes the paper hard to read.

5. Although the case analysis in section 4 is interesting, the result is only applicable to one very specific data distribution. Can the authors connect the data distribution to more general cases?

Minor

1. Missing section number of "RELATED WORK"

2. Why do the authors use ridge regression to measure the expressive power instead of plain linear regression without regularziation?

**Questions:**

See "Weakness" section.

---

> ### Author Response · Authors · 2023-11-22
>
> We thank Reviewer 6Vs9 for their comments. We address their concerns as follows:
>
> **"The true expressive power should be analyzed in terms of the function classes of these two kinds of networks."**\
> In this paper (Section 2.2), the expressivity of a model follows from the set of functions that it can express through the linear combination of its features (that are fixed as they are learned features after training), which is different from any functions the model can express in theory without considering the optimization (learning) part. The true expressive power should be analyzed in terms of the function classes of these two kinds of networks.Optimization bias  may force models of different size to pick different features.
>
> **"The paper does not justify why comparing overparameterized models with an ensemble of shallow models is important or meaningful. As an ensemble of small networks has fewer parameters than the large network, why is this a fair comparison?"**\
> We include enough small networks in the ensemble to match the total number of parameters present in the large network for a fair comparison.
>
> **"Although the case analysis in section 4 is interesting, the result is only applicable to one very specific data distribution. Can the authors connect the data distribution to more general cases?"**\
> Section 4 aims to give insights into why we observe the results in the previous sections. Some assumptions are needed to illustrate these observations in a simpler setting.

---

> ### Comment · Reviewer_6Vs9 · 2023-11-23
> **Post-rebuttal comments**
>
> Thanks for the response from the authors. After reading the response, I decide to keep the score unchanged.

---

### Official Review · Reviewer_3arT · 2023-10-31

**Soundness:** 4 excellent
**Presentation:** 4 excellent
**Contribution:** 2 fair
**Rating:** 6
**Confidence:** 4

**Summary:**

They study the neural representations of thin and wide deep neural networks. Their main finding is that concatenating the latent representations of multiple thin networks does not result in representations that are as useful as a single wide neural network. Their primary experiments involve seeing how well the activations of a wide network can be reconstructed from the concatenated activations of thin networks using a linear layer and vice versa.

**Strengths:**

- I think that the approach is mostly novel and clever. I think the results make a very clear case for their conclusions. This seems like a valuable piece of evidence related to understanding neural representations. I’m glad this work was done.
- The paper is very well-written.

**Weaknesses:**

1. I think this paper is well-done but lags somewhat behind its time. I think this area of research was much more popular and cutting-edge a few years ago. In that sense, I think this paper can be a good one but probably is not groundbreaking enough to be great. This criticism will not factor into my overall rating.
2. The experiments did not scale past the CIFAR and MNLI scale.
3. I think there are some related works that should have been discussed. I recommend considering adding the ones below.
    - https://arxiv.org/abs/2212.11005
    - https://arxiv.org/abs/2106.07682
    - https://arxiv.org/abs/2110.14633
    - https://arxiv.org/abs/2010.02323
    - https://arxiv.org/abs/1912.04783
4. My biggest reservation about the paper is that there are multiple ways of comparing the similarity of neural representations. This paper introduces the FSE and FSG, but I do not see why prior methods were not considered. At a minimum, these deserve discussion. Section V.G of [Rauker et al. (2022)](https://arxiv.org/abs/2207.13243) discusses single neuron alignment, vector space alignment, CCA, singular vector CCA, CKA, deconfounded representation similarity, layer reconstruction, model stitching, representational similarity analysis, and probing. I do not think that the paper does a good job of overviewing related work and comparing their measures against baselines.

**Questions:**

5. Why use a linear layer to define the FSE? Why not allow yourself to use a nonlinear layer? Other works from the model stitching literature have done this, e.g. [Bansal et al. (2021)](https://arxiv.org/abs/2106.07682). I would not be shocked if the main result from 3.2 didn’t hold much for a nonlinear version of FSE.
6 I see no error bars in some of the figures. Were these results based on one trial? Or are the error bars too small to see?

---

> ### Author Response · Authors · 2023-11-22
>
> We thank Reviewer 3arT for their comments. We address their concerns as follows:
>
> **"The experiments did not scale past the CIFAR and MNLI scale."**\
> We preferred smaller scale experiments because for our results we solve many regression problems that becomes computationally very expensive as the dataset and the networks grow.
>
> **"My biggest reservation about the paper is that there are multiple ways of comparing the similarity of neural representations."**\
> As we state in our introduction, our focus (with FSE and FSG) is not on similarity  rather the expressivity that is dictated by the features in last layer of a network. The similarity between A and B  is the same as similarity between B and A but that is not the case for expressivity. FSE and FSG help us understand the expressive power of the features of different size networks relative to each other and quality of the feature on a given task.
>
> **"Why use a linear layer to define the FSE? Why not allow yourself to use a nonlinear layer?"**\
> As explained in Section 2.2, the model makes predictions based on the linear combination of its (last layer) features - each output neuron (before softmax) is linear combination of these features. Hence, the expressive power of a model is dictated by the linear span of the last hidden layer features (that is only the case for the last hidden layer), and this expressive power does not depend on non-linear relationships between the last hidden layer features. This expressive power's dependence on the linear combination of these features is the basis of the FSE measure, which quantifies one model's expressive capacity with respect to another.
>
> **"I see no error bars in some of the figures. Were these results based on one trial? Or are the error bars too small to see?"**\
> Error bars are too small to see in most cases. Experiments are averaged over 5 trials.

---

> > ### Comment · Reviewer_3arT · 2023-11-23
> > **Re: 2-3**
> >
> > Thanks for the clarification and help. It seems that I missed the part saying that only the last layer is used. But this seems to itself be a big limitation of what the paper does, right? I would imagine that we should care about others as well. Other methods of comparing representational similarities of which I know have been applied in a more versatile way.

---

> > > ### Author Response · Authors · 2023-11-23
> > >
> > > We don't think it is a limitation that we focus on the last layer features. The expressive power of a model depends the linear span of its last layer features. Hence, a direct comparison of last layer features of two models help us fully understand these two models' expressivity relative to each other. We cannot have the same insights by comparing the intermediate layer features only.
> > > \
> > > If  last layer features of model A can span those of model B perfect, i.e. FSE score is zero, that means model A can express any function model B can express. We cannot have the same conclusion by comparing the intermediate layers.

---

### Meta-Review · Area_Chair_MMHi · 2023-12-13

**Metareview:**

This paper attempts to understand how overparameterization affects the features learned by the neural networks. To understand this, the authors compare the features learned by a concatenation of under-parameterized models and an over-parameterized model. The authors show that the concatenation of under-parameterized models cannot span the space of features learned by the overparameterized model.

There were several questions raised by the reviewers. The authors gave convincing replies to most of the questions. Particularly, I am not rejecting this paper for simplicity of tasks or lack of novelty of the similarity metric when compared to existing metrics.

There were two valid criticisms that are worth considering:

First, reviewer Si1X mentioned that an alternative explanation for the results could be just that the under-parameterized models have high error and hence poorly trained. While authors argue that they are not poorly trained, both versions of the models do not have the same accuracy so this needs an explanation.

My main reason for rejecting the paper is the second criticism by the reviewer Y4Do who questions the set-up itself since a bunch of under-parameterized models could actually just learn the same or similar representation and hence they cannot learn the same representation as the over-parameterized models. This could be mostly true because these small models are trained independently. There is no good answer from the authors to this question.

I suggest the authors address these 2 concerns and resubmit.

**Justification For Why Not Higher Score:**

There are 2 valid criticisms and one of them actually questions the basis of this paper.

**Justification For Why Not Lower Score:**

N/A

---

### Decision · Program_Chairs · 2024-01-16

Reject